# Manifold Topology Divergence: a Framework for Comparing Data Manifolds

**Serguei Barannikov**
Skolkovo Institute of Science and Technology
Moscow, Russia
CNRS, IMJ, Paris University, France

**Ilya Trofimov**
Skolkovo Institute of Science and Technology
Moscow, Russia

**Grigorii Sotnikov**
Skolkovo Institute of Science and Technology
Moscow, Russia

**Ekaterina Trimbach**
Moscow Institute of Physics and Technology
Moscow, Russia

**Alexander Korotin**
Skolkovo Institute of Science and Technology,
Artificial Intelligence Research Institute (AIRI),
Moscow, Russia

**Alexander Filippov**
Huawei Noah's Ark Lab

**Evgeny Burnaev**
Skolkovo Institute of Science and Technology,
Artificial Intelligence Research Institute (AIRI),
Moscow, Russia

## Abstract

We develop a framework for comparing data manifolds, aimed, in particular, towards the evaluation of deep generative models. We describe a novel tool, Cross-Barcode(P,Q), that, given a pair of distributions in a high-dimensional space, tracks multiscale topology spacial discrepancies between manifolds on which the distributions are concentrated. Based on the Cross-Barcode, we introduce the Manifold Topology Divergence score (MTop-Divergence) and apply it to assess the performance of deep generative models in various domains: images, 3D-shapes, time-series, and on different datasets: MNIST, Fashion MNIST, SVHN, CIFAR10, FFHQ, chest X-ray images, market stock data, ShapeNet. We demonstrate that the MTop-Divergence accurately detects various degrees of mode-dropping, intra-mode collapse, mode invention, and image disturbance. Our algorithm scales well (essentially linearly) with the increase of the dimension of the ambient high-dimensional space. It is one of the first TDA-based practical methodologies that can be applied universally to datasets of different sizes and dimensions, including the ones on which the most recent GANs in the visual domain are trained. The proposed method is domain agnostic and does not rely on pre-trained networks.

## 1 Introduction

Geometric perspective in working with data distributions has been pervasive in machine learning [5, 8, 12, 10, 23, 20]. Reconstruction of the data from observing only a subset of its points has made a significant step forward since the invention of Generative Adversarial Networks (GANs) [13].

35th Conference on Neural Information Processing Systems (NeurIPS 2021).

Despite the exceptional success that deep generative models achieved, there still exists a longstanding challenge of good assessment of the generated samples quality and diversity [7].

For images, the Fréchet Inception Distance (FID) [16] is the most popular GAN evaluation measure. However, FID is limited only to 2D images since it relies on pre-trained on ImageNet "Inception" network. FID unrealistically approximates point clouds by Gaussians in embedding space; also, FID is biased [6]. Surprisingly, FID can't be applied to compare adversarial and non-adversarial generative models since it is overly pessimistic to the latter ones [28].

The evaluation of generative models is about comparing two *point clouds*: the true data cloud $P_{\text{data}}$ and the model (generated) cloud $Q_{\text{model}}$. In view of the commonly assumed Manifold Hypothesis [5, 12], we develop a topology-based measure for comparing two manifolds: the true data manifold $M_{\text{data}}$ and the model manifold $M_{\text{model}}$, by analysing samples $P_{\text{data}} \subset M_{\text{data}}$ and $Q_{\text{model}} \subset M_{\text{model}}$.

**Contribution.** In this work, we make the following contributions:

1. We introduce a new tool: Cross-Barcode$(P, Q)$. For a pair of point clouds $P$ and $Q$, the Cross-Barcode$(P, Q)$ records the differences in multiscale topology between two manifolds approximated by the point clouds;
2. We propose a new measure for comparing two data manifolds approximated by point clouds: Manifold Topology Divergence (MTop-Div);
3. We apply the MTop-Div to evaluate performance of GANs in various domains: 2D images, 3D shapes, time-series. We show that the MTop-Div correlates well with domain-specific measures and can be used for model selection. Also it provides insights about evolution of generated data manifold during training;
4. We have compared the MTop-Div against 7 established evaluation methods: FID, discriminative score, MMD, JSD, 1-coverage, IMD and Geometry score and found that MTop-Div is able to capture subtle differences in data geometry;
5. We have essentially overcame the known TDA scalability issues and in particular have carried out the MTop-Div calculations on most recent datasets such as FFHQ, with dimensions $D$ up to $10^7$.

The source code is available at `https://github.com/IlyaTrofimov/MTopDiv`.

**Related work.** GANs try to recover the true data distribution via a min-max game where two players, typically represented by deep neural networks, called Discriminator and Generator, compete by optimizing the common objective. Training curves are not informative since generator and discriminator counter each other, thus, the loss values are often meaningless. Several other measures were introduced to estimate the quality of GANs. However, there is no consensus which of them best captures strengths and limitations of the models and can be used for the fair model selection. The likelihood in Parzen window works poorly in high-dimensional spaces and does not correlate with the visual quality of generated samples [30]. The Inception Score (IS) [29] measures both the discriminability and diversity of generated samples. It relies on the pre-trained Inception network and is limited to 2D images domain. While IS correlates with visual image quality, it ignores the true data distribution, doesn't detect mode dropping and is sensitive to image resolution. The Fréchet Inception Distance (FID) [16] is a distance between two multivariate Gaussians. These Gaussians approximate the features of generated and true data extracted from the last hidden layer of the pretrained Inception network. Similar variant is KID [6] which computes MMD distance between two distributions of features. The work [14] proposed to use the Duality Gap, a notion from the game theory, as a domain agnostic measure for GAN evaluation. Another variant of a measure inspired by game theory was proposed in [27]. Conventional precision and recall measures can be also used [22, 21]. Accurate calculation of precision and recall is limited to simple datasets from low-dimensional manifolds. The *Geometry Score* (GScore) [19] is the L2-distance between mean Relative Living Times (RLT) of topological features calculated for the model distribution and the true data distribution. The GScore is domain agnostic, does not involve auxiliary pretrained networks and is not limited to 2D images. However, GScore is not sensitive even to some simple transformations - like constant shift, dilation, or reflection (see our Fig. 2, 4). The barcodes in GScore are calculated approximately, based only on the approximate witness complexes on 64 landmark points sampled from each distribution. That's why the procedure is stochastic and should be repeated several thousand times for averaging. Thus, the calculation of GScore can be prohibitively long for large datasets. We also refer reader to the comprehensive survey [7].

## 2 Cross-Barcode and Manifold Topology Divergence

### 2.1 Multiscale simplicial approximation of manifolds

According to the well-known Manifold Hypothesis [12] the support of the data distribution $\mathcal{P}_{\text{data}}$ is often concentrated on a low-dimensional manifold $M_{\text{data}}$. We construct a framework for comparing numerically such distribution $\mathcal{P}_{\text{data}}$ with a similar distribution $\mathcal{Q}_{\text{model}}$ concentrated on a manifold $M_{\text{model}}$. Such distribution $\mathcal{Q}_{\text{model}}$ is produced, for example, by a generative deep neural network in one of applications' scenarios. The immediate difficulty here is that the manifold $M_{\text{data}}$ is unknown and is described only through discrete sets of samples from the distribution $\mathcal{P}_{\text{data}}$. One standard approach to resolve this difficulty is to approximate the manifold $M_{\text{data}}$ by simplices with vertices given by the sampled points. The simplices approximating the manifold are picked based on proximity information given by the pairwise distances between sampled points [5, 25]. The standard approach is to fix a threshold $r > 0$ and to take the simplices whose edges do not exceed the threshold $r$. The choice of threshold is essential here since if it is too small, then only the initial points, i.e., separated from each other 0-dimensional simplices, are allowed. And if the threshold is too large, then all possible simplices with sampled points as vertices are included and their union is simply the big blob representing the convex hull of the sampled points. Instead of trying to guess the right value of the threshold, the standard recent approach is to study all thresholds at once. This can be achieved thanks to the mathematical tool, called barcode [2, 10], that quantifies the evolution of topological features over multiple scales. For each value of $r$ the barcode describes the topology, namely the numbers of holes or voids of different dimensions, of the union of all simplices included up to the threshold $r$.

### 2.2 Measuring the differences in simplicial approximation of two manifolds

However, to estimate numerically the degree of similarity between the manifolds $M_{\text{model}}, M_{\text{data}} \subset \mathbb{R}^D$, it is important not just to know the numbers of topological features across different scales for simplicial complexes approximating $M_{\text{model}}, M_{\text{data}}$, but to be able to verify that the similar topological features are located at similar places and appear at similar scales.

Our method measures the differences in the simplicial approximation of the two manifolds, represented by samples $P$ and $Q$, by constructing sets of simplices, describing discrepancies between the two manifolds. To construct these sets of simplices we take the edges connecting $P-$points with $Q-$points, and also $P-$points between them, ordered by their length, and start adding these edges one by one, beginning from the smallest edge and gradually increasing the threshold, see Figure 1. We add also the triangles and $k-$simplices at the threshold when all their edges have been added. It is assumed that all edges between $Q-$points were already in the initial set. We track in this process the births and the deaths of topological features, where the topological features are allowed

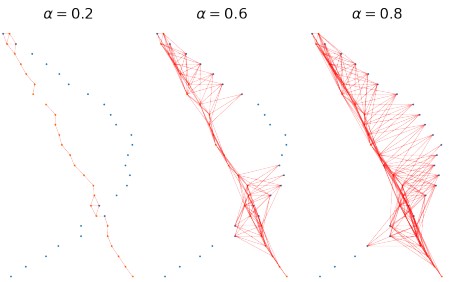

Figure 1: Edges(red) connecting $P-$points(red) with $Q-$points(blue), and also $P-$points between them, are added for three thresholds: $\alpha = 0.2, 0.4, 0.6$

here to have boundaries on any simplices formed by $Q-$points. The longer the lifespan of the topological feature across the change of threshold the bigger the described by this feature discrepancy between the two manifolds.

**Homology** is a tool that permits to single out topological features that are similar, and to decompose any topological feature into a sum of basic topological features. More specifically, in our case, a $k-$cycle is a collection of $k-$simplices formed by $P-$ and $Q-$ points, such that their boundaries cancel each other, perhaps except for the boundary simplices formed only by $Q-$points. For example, a cycle of dimension $k = 1$ corresponds to a path connecting a pair of $Q-$points and consisting of edges passing through a set of $P-$points. A cycle which is a boundary of a set of $(k + 1)-$simplices is considered trivial. Two cycles are topologically equivalent if they differ by a boundary, and by collection of simplices formed only by $Q-$points. A union of cycles is again a cycle. Each cycle can be represented by a vector in the vector space where each simplex corresponds to a generator. In practice, the vector space over $\{0, 1\}$ is used most often. The union of cycles corresponds to the sum of vectors. The homology vector space $H_k$ is defined as the factor of the vector space of all $k-$cycles

modulo the vector space of boundaries and cycles consisting of simplices formed only by $Q-$points. A set of vectors forming a basis in this factor-space corresponds to a set of basic topological features, so that any other topological feature is equivalent to a sum of some basic features.

The homology are also defined for manifolds and for arbitrary topological spaces. This definition is technical and we have to omit it due to limited space, and to refer to e.g. [15, 24] for details. The relevant properties for us are the following. For each topological space $X$ the vector spaces $H_k(X)$, $k = 0, 1 \ldots$, are defined. The dimension of the vector space $H_k$ equals to the number of independent $k-$dimensional topological features (holes,voids etc). An inclusion $Y \subset X$ induces a natural map $H_k(Y) \rightarrow H_k(X)$

In terms of homology, we would like to verify that not just the dimensions of homology groups $H_*(M_{\text{model}})$ and $H_*(M_{\text{data}})$ are the same but that more importantly the natural maps:

$$\varphi_r : H_*(M_{\text{model}} \cap M_{\text{data}}) \rightarrow H_*(M_{\text{model}}) \tag{1}$$

$$\varphi_p : H_*(M_{\text{model}} \cap M_{\text{data}}) \rightarrow H_*(M_{\text{data}}) \tag{2}$$

induced by the embeddings are as close as possible to isomorphisms. The homology of a pair is the tool that measures how far such maps are from isomorphisms. Given a pair of topological spaces $Y \subset X$, the homology of a pair $H_*(X, Y)$ counts the number of independent topological features in $X$ that cannot be deformed to a topological feauture in $Y$ plus independent topological features in $Y$ that, after the embedding to $X$, become deformable to a point. An equivalent description, the homology of a pair $H_*(X, Y)$ counts the number of independent topological features in the factor-space $X/Y$, where all points of $Y$ are contracted to a single point. The important fact for us is that the map, induced by the embedding, $H_*(Y) \rightarrow H_*(X)$ is an isomorphism if and only if the homology of the pair $H_*(X, Y)$ are trivial. Moreover the embedding of simple simplicial complexes $Y \subset X$ is an equivalence in homotopy category, if and only if the homology of the pair $H_*(X, Y)$ are trivial [33].

To define the counterpart of this construction for manifolds represented by point clouds, we employ the following strategy. Firstly, we replace the pair $(M_{\text{model}} \cap M_{\text{data}}) \subset M_{\text{model}}$ by the equivalent pair $M_{\text{model}} \subset (M_{\text{data}} \cup M_{\text{model}})$ with the same factor-space. Then, we represent $(M_{\text{data}} \cup M_{\text{model}})$ by the union of point clouds $P \cup Q$, where the point clouds $P, Q$ are sampled from the distributions $\mathcal{P}_{\text{data}}$, $\mathcal{Q}_{\text{model}}$. Our principal claim here is that taking topologically the quotient of $(M_{\text{data}} \cup M_{\text{model}})$ by $M_{\text{model}}$ is equivalent in the framework of multiscale analysis of topological features to the following operation on the matrix $m_{P \cup Q}$ of pairwise distances of the cloud $P \cup Q$: **we set to zero all pairwise distances within the subcloud $\mathbf{Q} \subset (P \cup Q)$.**

## 2.3 Cross-Barcode (P,Q)

Let $P = \{p_i\}$, $Q = \{q_j\}$, $p_i, q_j \in \mathbb{R}^D$ are two point clouds sampled from two distributions $\mathcal{P}$, $\mathcal{Q}$. To define Cross-Barcode$(P, Q)$ we construct first the following filtered simplicial complex. Let $(\Gamma_{P \cup Q}, m_{(P \cup Q)/Q})$ be the weighted graph with the distance-like weights on edges defined as the complete graph on the union of point clouds $P \cup Q$ with the distance matrix given by the pairwise distance in $\mathbb{R}^D$ for the pairs of points $(p_i, p_j)$ or $(p_i, q_j)$ and with all pairwise distances within the cloud $Q$ that we set to zero. Our filtered simplicial complex is the Vietoris-Rips complex of $(\Gamma_{P \cup Q}, m_{(P \cup Q)/Q})$.

Recall that given such a graph $\Gamma$ with matrix $m$ of pairwise distances between vertices and a parameter $\alpha > 0$, the Vietoris-Rips complex $R_\alpha(\Gamma, m)$ is the abstract simplicial complex with simplices that correspond to the non-empty subsets of vertices of $\Gamma$ whose pairwise distances are less than $\alpha$ as measured by $m$. Increasing parameter $\alpha$ adds more simplices and this gives a nested family of collections of simplices know as filtered simplicial complex. Recall that a **simplicial complex** is described by a set of vertices $V = \{v_1, \ldots, v_N\}$, and a collection of $k-$simplices $S$, i.e. $(k + 1)-$elements subsets of the set of vertices $V$, $k \geq 0$. The set of simplices $S$ should satisfy the condition that for each simplex $s \in S$ all the $(k - 1)$-simplices obtained by the deletion of a vertex from the subset of vertices of $s$ belong also to $S$. The **filtered simplicial complexes** is the family of simplicial complexes $S_\alpha$ with nested collections of simplices: for $\alpha_1 < \alpha_2$ all simplices of $S_{\alpha_1}$ are also in $S_{\alpha_2}$.

At the initial moment, $\alpha = 0$, the simplicial complex $R_\alpha(\Gamma_{P \cup Q}, m_{(P \cup Q)/Q})$ has trivial homology $H_k$ for all $k > 0$ since it contains all simplices formed by $Q-$points. The dimension of the $0-$th

homology equals at $\alpha = 0$ to the number of $P-$points, since no edge between them or between a $P-$point and a $Q-$point is added at the beginning. As we increase $\alpha$, some cycles, holes or voids appear in our complex $R_\alpha$. Then, some combinations of these cycles disappear. The **persistent homology** principal theorem [2, 37] implies that it is possible to choose the set of generators in the homology of filtered complexes $H_k(R_\alpha)$ across all the scales $\alpha$ such that each generator appears at its specific "birth" time and disappears at its specific "death" time. These sets of "birth" and "deaths" of topological features in $R_\alpha$ are registered in **Barcode** of the filtered complex.

**Definition.** The Cross-Barcode$_i(P, Q)$ is the set of intervals recording the "births" and "deaths" times of $i-$dimensional topological features in the filtered simplicial complex $R_\alpha(\Gamma_{P \cup Q}, m_{(P \cup Q)/Q})$.

Examples of Cross-Barcode$_i(P, Q)$ are shown on Fig. 2, 11, 14, 17, 18. Topological features with longer "lifespan" are considered essential. The topological features with "birth"="death" are trivial by definition and do not appear in Cross-Barcode$_*(P, Q)$.

### 2.4 Basic properties of Cross-Barcode$_*$(P,Q)

**Proposition 1.** Here is a list of basic properties of Cross-Barcode$_*(P, Q)$:

- if the two clouds coincide then Cross-Barcode$_*(P, P) = \varnothing$;
- for $Q = \varnothing$, Cross-Barcode$_*(P, \varnothing) = $ Barcode$_*(P)$, the barcode of the single point cloud $P$ itself;
- the norm of Cross-Barcode$_i(P, Q)$, $i \geq 0$, is bounded from above by the Hausdorff distance

$$\|\text{Cross-Barcode}_i (P, Q)\|_B \leq d_H(P, Q). \tag{3}$$

The proof is given in appendix.

### 2.5 The Manifold Topology Divergence (MTop-Div)

The bound from eq.(3) and the equality Cross-Barcode$_*(P, P) = \varnothing$ imply that the closeness of Cross-Barcode$_*(P, Q)$ to the empty set is a natural measure of discrepancy between $\mathcal{P}$ and $\mathcal{Q}$. Each Cross-Barcode$_i(P, Q)$ is a list of intervals describing the persistent homology $H_i$. To measure the closeness to the empty set, one can use segments' statistics: sum of lengths, sum of squared lengths, number of segments, the maximal length (H$_i$ max) or specific quantile. We assume that various characteristics of different $H_i$ could be useful in various cases, but the cross-barcodes for $H_0$ and $H_1$ can be calculated relatively fast.

Our **MTop-Divergence**$(\mathcal{P}, \mathcal{Q})$ is based on the sum of lengths of segments in Cross-Barcode$_1(P, Q)$, see section 2.6 for details. The sum of lengths of segments in Cross-Barcode$_1(P, Q)$ has an interesting interpretation via the Earth Mover's Distance. Namely, it is easy to prove (see Appendix B.2) that EM-Distance between the Relative Living Time histogram for Cross-Barcode$_1(P, Q)$ and the histogram of the empty barcode, multiplied by the parameter $\alpha_{max}$ from the definition of RLT, see e.g. [19], coincides with the sum of lengths of segments in $H_1$. This ensures the standard good stability properties of this quantity.

Our metrics can be applied in two settings: to a pair of distributions $\mathcal{P}_{\text{data}}, \mathcal{Q}_{\text{model}}$, in which case we denote our score MTop-Div(D,M), and to a pair of distributions $\mathcal{Q}_{\text{model}}, \mathcal{P}_{\text{data}}$, in which case our score is denoted MTop-Div(M,D). These two variants of the Cross-Barcode, and of the MTop-Divergence are related to the concepts of precision and recall. These two variants can be analyzed separately or combined together, e.g. averaged.

### 2.6 Algorithm

To calculate the score that evaluates the similitude between two distributions, we employ the following algorithm. First, we compute Cross-Barcode$_1(P, Q)$ on point clouds $P, Q$ of sizes $b_P, b_Q$ sampled from the two distributions $\mathcal{P}, \mathcal{Q}$. For this we calculate the matrices $m_P, m_{P,Q}$ of pairwise distances within the cloud $P$ and between clouds $P$ and $Q$. Then the algorithm constructs the Vietoris-Rips filtered simplicial complex from the matrix $m_{(P \cup Q)/Q}$ which is the matrix of pairwise distances in $P \cup Q$ with the pairs of points from cloud $Q$ block replaced by zeroes and with other blocks given by $m_P, m_{P,Q}$. Next step is to calculate the barcode of the constructed filtered simplicial complex. This step and the previous step constructing the filtered complex from the matrix $m_{(P \cup Q)/Q}$ can

**Algorithm 1** Cross-Barcode$_i(P, Q)$

**Input:** $m[P,P], m[P,Q]$ : matrices of pairwise distances within point cloud $P$, and between point clouds $P$ and $Q$

**Require:** VR($M$): function computing filtered complex from pairwise distances matrix $M$

**Require:** B($C, i$): function computing persistence intervals of filtered complex $C$ in dimension $i$

$b_Q \leftarrow$ number of columns in matrix $m[P,Q]$

$m[Q,Q] \leftarrow$ zeroes$(b_Q, b_Q)$

$M \leftarrow \begin{pmatrix} m[P,P] & m[P,Q] \\ m[P,Q] & m[Q,Q] \end{pmatrix}$

Cross-Barcode$_i \leftarrow$ B(VR($M$), $i$)

**Return:** list of intervals **Cross-Barcode**$_i(P, Q)$ representing "births" and "deaths" of topological discrepancies

---

**Algorithm 2** MTop-Divergence($\mathcal{P}, \mathcal{Q}$), see section 2.6 for details, default suggested values: $b_{\mathcal{P}} = 1000, b_{\mathcal{Q}} = 10000, n = 100$

**Input:** $X_{\mathcal{P}}, X_{\mathcal{Q}}$: $N_{\mathcal{P}} \times D, N_{\mathcal{Q}} \times D$ arrays representing datasets

  **for** $j = 1$ **to** $n$ **do**

    $P_j \leftarrow$ random choice$(X_{\mathcal{P}}, b_{\mathcal{P}})$

    $Q_j \leftarrow$ random choice$(X_{\mathcal{Q}}, b_{\mathcal{Q}})$

    $\mathcal{B}_j \leftarrow$ list of intervals Cross-Barcode$_1(P_j, Q_j)$ calculated by Algorithm1

    $mtd_j \leftarrow$ sum of lengths of all intervals in $\mathcal{B}_j$

  **end for**

MTop-Divergence($\mathcal{P}, \mathcal{Q}$) $\leftarrow$ mean($mtd$)

**Return:** number **MTop-Divergence**($\mathcal{P}, \mathcal{Q}$) representing discrepancy between the distributions $\mathcal{P}, \mathcal{Q}$

---

be done using one of the fast scripts[1], some of them are optimized for GPU acceleration, see e.g. [35, 4]. The calculation of barcode from the filtered complex is based on the persistence algorithm bringing the filtered complex to its "canonical form" ([2]). Next, sum of lengths or one of other numerical characteristcs of Cross-Barcode$_1(P, Q)$ is computed. Then this computation is run a sufficient number of times to obtain the mean value of the picked characteristic. In our experiments we have found that for common datasets the number of times from 10 to 100 is generally sufficient. Our method is summarized in the Algorithms 1 and 2.

**Complexity.** The Algorithm 1 requires computation of the two matrices of pairwise distances $m[P,P]$, $m[P,Q]$ for a pair of samples $P \in \mathbb{R}^{b_{\mathcal{P}} \times D}$, $Q \in \mathbb{R}^{b_{\mathcal{Q}} \times D}$ involving $O(b_{\mathcal{P}}^2 D)$ and $O(b_{\mathcal{P}} b_{\mathcal{Q}} D)$ operations. After that, the complexity of the computation of barcode does not depend on the dimension $D$ of the data. Generally the persistence algorithm is at worst cubic in the number of simplices involved. In practice, the boundary matrix is sparse in our case and thanks also to the GPU optimization, the computation of cross-barcode takes similar time as in the previous step on datasets of big dimensionality. Since only the discrepancies in manifold topology are calculated, the results are quite robust and a relatively low number of iterations is needed to obtain accurate results. Since the algorithm scales linearly with $D$ it can be applied to the most recent datasets with $D$ up to $10^7$. For example, for $D = 3.15 \times 10^6$, and batch sizes $b_{\mathcal{P}} = 10^3, b_{\mathcal{Q}} = 10^4$, on NVIDIA TITAN RTX the time for GPU accelerated calculation of pairwise distances was 15 seconds, and GPU-accelerated calculation of Cross-Barcode$_1(P, Q)$ took 30 seconds.

## 3 Experiments

We examine the ability of MTop-Div to measure quality of generative models trained on various datasets. Firstly, we illustrate the behaviour of MTop-Div on simple synthetic datasets (rings (Fig.2), disks (Fig. 13)). Secondly, we show that MTop-Div is superior to the GScore, another topology-based GAN quality measure. We carry out experiments with a mixture of Gaussians, MNIST, CIFAR10, X-ray images, FFHQ. The performance of MTop-Div is on par with FID. For images, MTop-Div is always calculated in pixel space without utilizing pre-trained networks. Thirdly, we apply MTop-Div to GANs from non-image domains: 3D shapes and time-series, where FID is not applicable. We show that MTop-Div agrees with domain-specific measures, such as JSD, MMD, Coverage, discriminative score, but MTop-Div better captures evolution of generated data manifold during training [2].

### 3.1 Simple synthetic datasets in 2D

As illustrated on Fig. 2 the GScore does not respond to shifts of the distributions' relative position.

---

[1]Persistent Homology Computation (wiki)

[2]we additionally calculated IMD [31] for the pairs of point clouds from our experiments, see Appendix E.

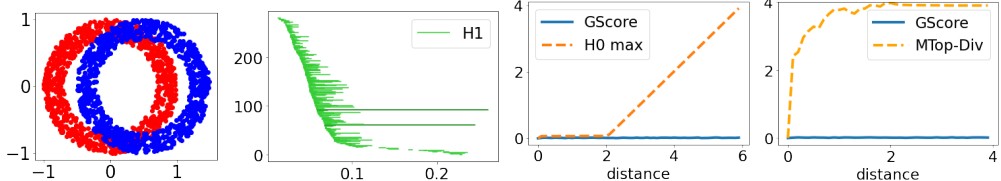

Figure 2: MTop-Div and H0 max compared with GScore, for two ring clouds of 1000 points, as function of $d$ =distance between ring centers, the Cross-Barcode$_1(P,Q)$ is shown at $d = 0.5$

## 3.2 Mode-Droping on Gaussians

One of common tasks to assess GAN's performance is measuring it's ability to uncover the variety of modes in the data distribution. A common benchmark for this problem is a mixture of Gaussians, see Fig. 3. We trained two generators with very different performance: original GAN, which managed to capture all 5 modes of the distribution and WGAN-GP, which have only covered poorly two. However, the Geometry score is not sensitive to such a difference since two point clouds have the same RLT histogram. While the MTop-Div is sensitive to such a difference.

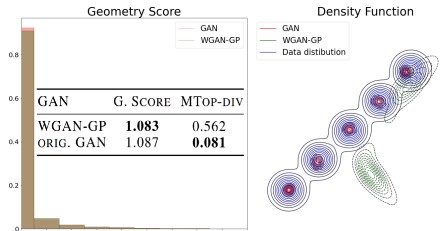

Figure 3: Difficulty of the Geometry Score to detect the mode dropping.

## 3.3 Digit flipping on MNIST

Figure 4 shows an experiment with MNIST dataset. We compare two point clouds: "5"s vs. vertically flipped "5"s (resembling rather "2"s). These two clouds are indistinguishable for Geometry Score, while the MTop-Div is sensitive to such flip since it depends on the relative position of the two clouds.

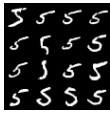 vs 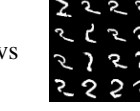

Geometry Score = 0.0
MTop-Div = 6154.0

Figure 4: Two point clouds:"5"s from MNIST vs. vertically flipped "5"s from MNIST (resembling rather "2"s). The two clouds are indistinguishable for Geometry Score, while the MTop-Div is sensitive to such flip as it depends on the positions of clouds with respect to each other.

## 3.4 Synthetic modifications of CIFAR10

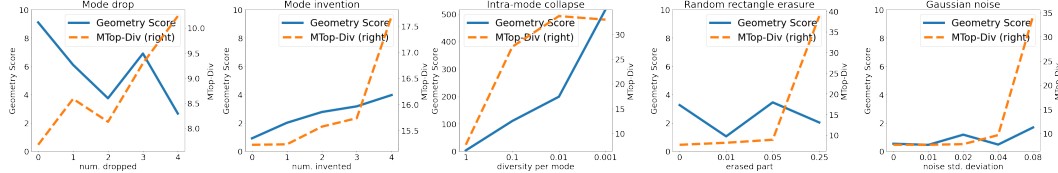

Figure 5: Experiment with modifications of CIFAR10. The disturbance level rises from zero to a maximum. Ideally, the quality score should monotonically increase with the disturbance level.

We evaluate the proposed MTop-Div(D,M) using a benchmark with the controllable disturbance level. We take CIFAR10 and apply the following modifications. Firstly, we emulate common issues in GAN: mode drop, mode invention and mode collapse by doing class drop, class addition and intra-class collapse (removal of objects within a class). Secondly, we apply two disturbance types: erasure of a random rectangle and a Gaussian noise. As 'real' images we use the test set from CIFAR10, as 'generated' images - a subsample of the train set with applied modifications. The size of the later always equals the test set size. Figure 5 shows the results. Ideally, the quality measure should monotonically increase with the disturbance level. We conclude that Geometry Score is monotone only for 'mode invention' and 'intra-mode collapse' while MTop-Div(D,M) is almost monotone for all the cases. The average Kendall-tau rank correlation between MTop-Div(D,M) and disturbance level is 0.89, while for Geometry Score the rank correlation is only 0.36. FID performs well on this

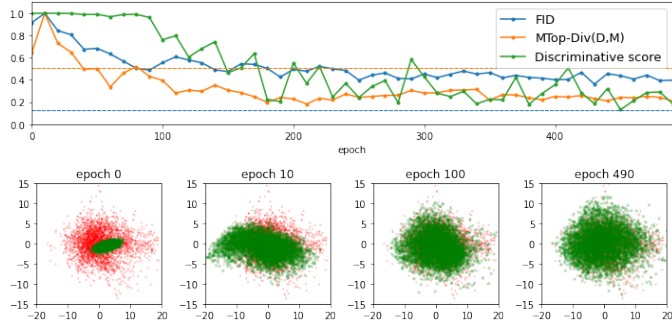

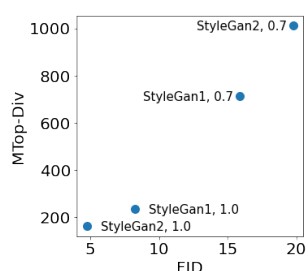

Figure 6: Training process of GAN applied to chest X-ray data. **Top**: normalized quality measures FID, MTop-Div, Disc. score vs. epoch. Each measure is divided by it's maximum value: max. FID = 304, max MTop-Div = 21, max. Disc.score = 0.5. Lower is better. Dashed horizontal lines depict comparison of real COVID-positive and COVID-neg. chest X-rays. **Bottom**: PCA projections of real objects (red) and generated objects (green).

Figure 7: Comparison of the quality measures, FID vs MTop-Div, on StyleGAN, StyleGAN2 trained on FFHQ with different truncation levels. MTop-Div(M,D) is monotonically increasing in good correlation with FID.

benchmark, not shown on Figure 5 for ease of perception. Additionally, we calculated MTop-Div for higher order Cross-Barcodes, see Appendix D.

### 3.5   GAN model selection

We trained WGAN and WGAN-GP on various datasets: CIFAR10, SVHN, MNIST, Fashion-MNIST and evaluated their quality, see Table 1. Experimental data show that the ranking between WGAN and WGAN-GP is consistent for FID and MTop-Div.

Table 1: MTop-Div is consistent with FID for model selection of GAN's trained on various datasets.

| Dataset | FID | | MTop-Div(M,D) | |
|---|---|---|---|---|
| | WGAN | WGAN-GP | WGAN | WGAN-GP |
| CIFAR10 | **154.6** | 399.2 | **353.1** | 1637.4 |
| SVHN | **101.6** | 154.7 | **332.0** | 963.2 |
| MNIST | 31.8 | **22.0** | 2042.8 | **1526.1** |
| FashionMNIST | 52.9 | **35.1** | 919.6 | **660.4** |

### 3.6   Experiments with StyleGAN

We evaluated the performance of StyleGAN [17] and StyleGAN2 [18] generators trained on the FFHQ dataset[3]. We generated $20 \times 10^3$ samples with two truncation levels: $\psi = 0.7, 1.0$ and compared them with $20 \times 10^3$ samples from FFHQ. The truncation trick is known to improve average image quality but decrease image variance. Figure 7 show the results (see also Table 2 in Appendix for more data). Thus, the ranking via MTop-Div(M, D) is consistent with FID. We also tried to calculate Geometry Score but found that it takes a prohibitively long time.

### 3.7   Chest X-rays generation for COVID-19 detection

Waheen et al. [32] described how to apply GANs to improve COVID-19 detection from chest X-ray images. Following [32], we trained an ACGAN [26] on a dataset consisting of chest X-rays of COVID-19 positive and healthy patients [4]. Next, we studied the training process of ACGAN. Every 10'th epoch we evaluated the performance of ACGAN by comparing real and generated COVID-19 positive chest X-ray images. That is, we calculated FID, MTop-Div(D,M) and a baseline measure -

---

[3]`https://github.com/NVlabs/ffhq-dataset` (CC-BY 2.0 License)

[4]We used the ACGAN implementation `https://github.com/clvrai/ACGAN-PyTorch`, (MIT License) and chest X-ray data was from `https://www.kaggle.com/tawsifurrahman/covid19-radiography-database` (Kaggle Data License)

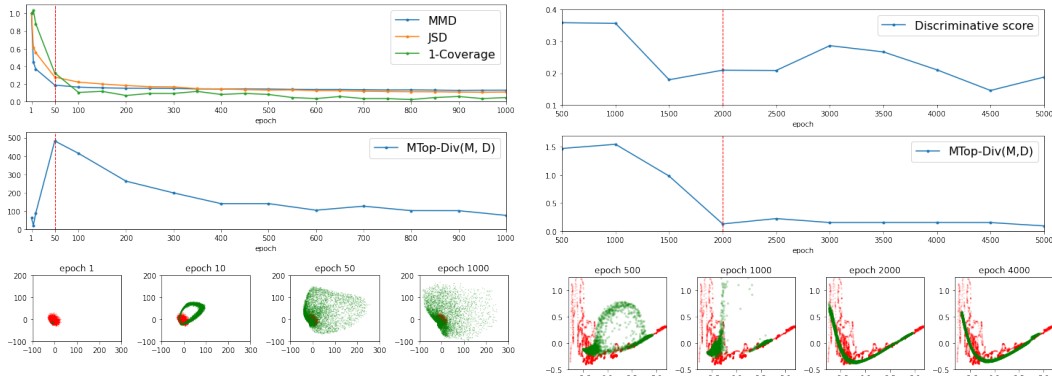

Figure 8: Training process of GAN applied to 3D shapes. **Top, middle**: quality measures MMD, JSD, 1-Coverage, MTop-Div vs. epoch. Each quality measure is normalized, that is, divided by its value at the first epoch. Lower is better. **Bottom**: PCA projection of real objects (red) and generated objects (green). **Vertical red line** (epoch 50) depicts the moment, when the manifold of generated objects "explodes" and becomes much more diverse.

Figure 9: Training dynamics of TimeGAN applied to market stock data. **Top**: discriminative score vs. epoch, MTop-Div vs. epoch. Lower is better. **Bottom**: PCA projection of real time-series (red) and generated time-series (green). **Vertical red line** (epoch 2000) depicts the moment when manifolds of real and generated objects become close.

discriminative score [5] of a CNN trained to distinguish real vs. generated data. The MTop-Div agrees with FID and the discriminative score. PCA projections show that generated data approximates real data well. Figure 20 in Appendix presents real and generated images.

Additionally, we compared real COVID-positive and COVID-negative chest X-ray images, see horizontal dashed lines at Fig. 6. Counterintuitively, for FID real COVID-positive images are closer to real COVID-negative ones than to generated COVID-positive images; probably because FID is overly sensitive to textures. At the same time, evaluation by MTop-Div is consistent.

### 3.8 3D shapes generation

We use the proposed MTop-Div score to analyze the training process of the GAN applied to 3D shapes, represented by 3D point clouds [1]. For training, we used 6778 objects of the "chair" class from ShapeNet [9]. We trained GAN for 1000 epochs and tracked the following standard quality measures: Minimum Matching Distance (MMD), Coverage, and Jensen-Shannon Divergence (JSD) between marginal distributions. To understand the training process in more details, we computed PCA decomposition of real and generated objects (Fig. 8, bottom). For computing PCA, each object (3D point cloud) was represented by a vector of point frequencies attached to the 3-dimensional $28^3$ grid. Figure 8, top, shows that conventional metrics (MMD, JSD, Coverage) doesn't represent the training process adequately. While these measures steadily improve, the set of generated objects dramatically changes. At epoch 50, the set of generated objects (green) "explodes" and becomes much more diverse, covering a much larger space than real objects (red). Conventional quality measures (MMD, JSD, Coverage) ignore this shift while MTop-Div has a peak at this point. Next, we evaluated the final quality of GAN by training a classifier to distinguish real and generated object. A simple MLP with 3 hidden layers showed accuracy 98%, indicating that the GAN poorly approximates the manifold of real objects. This result is consistent with MTop-Div: at epoch 1000 it is even larger than at epoch 1.

### 3.9 Time series generation

Next, we analyze training dynamics of TimeGAN [34] tailored to multivariate time-series generation. We followed the experimental protocol from [34] and used the daily historical market stocks data

---

[5]Discriminative score equals accuracy minus 0.5. MTop-Div better correlates with the discriminative score than FID: 0.75 vs. 0.66.

from 2004 to 2019, including as features the volume, high, low, opening, closing, and adjusted closing prices. The baseline evaluation measure is calculated via a classifer (RNN) trained to distinguish real and generated time-series. Particularly, the *discriminative score* equals to the accuracy of such a classifer minus 0.5. Fig. 9, top, shows the results. We conclude that the behaviour of MTop-Div is consistent with the discriminative score: both of them decrease during training. To illustrate training in more details we did PCA projections of real and generated time-series by flattening the time dimension (Fig. 9, bottom). At 2000-th epoch, the point clouds of real (red) and generated (green) time-series became close, which is captured by a drop of MTop-Div score. At the same time, discriminative score is not sensitive enough to this phenomena.

## 4 Conclusions

We have proposed a tool, Cross-Barcode$_*(P, Q)$, which records multiscale topology discrepancies between two data manifolds approximated by point clouds. Based on the Cross-Barcode$_*(P, Q)$, we have introduced the Manifold Topology Divergence and have applied it to evaluate the performance of GANs in various domains: 2D images, 3D shapes, time-series. We have shown that the MTop-Div correlates well with domain-specific measures and can be used for model selection. Also, it provides insights about the evolution of generated data manifold during training and can be used for early stopping. The MTop-Div score is domain agnostic and does not rely on pre-trained networks. We have compared the MTop-Div against 7 established evaluation methods: FID, discriminative score, MMD, JSD, 1-coverage, IMD, and Geometry score and found that MTop-Div outperforms many of them and captures well subtle differences in data manifolds. Our methodology permits to overcome the known TDA scalability issues and to carry out the MTop-Div calculations on the most recent datasets such as FFHQ, with the size of up to $10^5$ and the dimension of up to $10^7$.

**Acknowledgements**

Evgeny Burnaev acknowledges the support of the Ministry of Science and Higher Education grant No. 075-10-2021-068. Authors are thankful to Alexei Artemov for help with the 3D shapes experiment.

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
