# A    Simplicial Complexes, Cycles, Barcodes

## A.1    Background

The simplicial complex is a combinatorial data that can be thought of as a higher-dimensional generalization of a graph. Simplicial complex $S$ is a collection of $k-$simplices, which are finite $(k+1)-$elements subsets in a given set $V$, for $k \geq 0$. The collection of simplices $S$ must satisfy the condition that for each $\sigma \in S$, $\sigma' \subset \sigma$ implies $\sigma' \in S$. A simplicial complex consisting only of $0-$ and $1-$simplices is a graph.

Let $C_k(S)$ denotes the vector space over a field $F$ whose basis elements are $k-$simplices from $S$ with a choice of ordering of vertices up to an even permutation. In calculations it is most convenient to put $F = \mathbb{Z}_2$. The boundary linear operator $\partial_k : C_k(S) \to C_{k-1}(S)$ is defined on $\sigma = \{x_0, \ldots, x_k\}$ as

$$\partial_k \sigma = \sum_{j=0}^{k} (-1)^j \{x_0, \ldots, x_{j-1}, x_{j+1}, \ldots, x_k\}.$$

The $k-$th **homology** group $H_k(S)$ is defined as the vector space $\ker \partial_k / \operatorname{im} \partial_{k+1}$. The elements $c \in \ker \partial_k$ are called **cycles**. The elements $\tilde{c} \in \operatorname{im} \partial_{k+1}$ are called boundaries. The general elements $c' \in C_k(S)$ are called chains. The elements of $H_k(S)$ represent various $k-$dimensional topological features in $S$. A basis in $H_k(S)$ corresponds to a set of basic topological features.

Filtration on simplicial complex is defined as a family of simplicial complexes $S_\alpha$ with nested collections of simplices: for $\alpha_1 < \alpha_2$ all simplices of $S_{\alpha_1}$ are also in $S_{\alpha_2}$. In practical examples the indexes $\alpha$ run through a discrete set $\alpha_1 < \ldots < \alpha_{\max}$.

The inclusions $S_\alpha \subseteq S_\beta$ induce naturally the maps on the homology groups $H_k(S_\alpha) \to H_k(S_\beta)$. The evolution of the cycles through the nested family of simplicial complexes $S_\alpha$ is described by the barcodes. The persistent homology principal theorem [2, 36, 37] states that for each dimension there exists a choice of a set of basic topological features across all $S_\alpha$ so that each feature appears in $H_k(S_\alpha)$ at specific time $\alpha = b_j$ and disappears at specific time $\alpha = d_j$. The $H_i$ barcode of the filtered simplicial complex is the record of these times represented as the collection of segments $[b_j, d_j]$. The barcodes are defined and calculated through bringing the set of matrices of the boundary operators $\partial_k$ to the "Canonical Form" by a change of the basis in $C_k$ preserving the nested family $S_\alpha$ [2, 3].

Let $(\Gamma, m)$ be a weighted graph with distance-like weights, where $m$ is the symmetric matrix of the weights attached to the edges of the graph $\Gamma$. The Vietoris-Rips filtered simplicial complex of the weighted graph $R_\alpha(\Gamma, m)$, is defined as the nested collection of simplices:

$$R_\alpha(\Gamma, m) = \{\{x_0, \ldots, x_k\}, x_i \in \operatorname{Vert}(\Gamma) \| m(x_j, x_l) \leq \alpha\}$$

where $\operatorname{Vert}(\Gamma)$ is the set of vertices of the graph $\Gamma$. Even though such weighted graphs do not always come from a set of points in metric space, barcodes of weighted graphs have been successfully applied in many situations (networks, fmri, medical data, graph's classification etc).

## A.2    Simplices, describing discrepancies between the two manifolds

Here we gather more details on the construction of sets of simplices that describe discrepancies between two point clouds $P$ and $Q$ sampled from the two distributions $\mathcal{P}, \mathcal{Q}$. As we have described in section 2, our basic methodology is to add consecutively the edges between $P-$points and $Q-$points and between pairs of $P-$points. All edges between $Q-$points are added simultaneously at the beginning at the threshold $\alpha = 0$. The $PP$ and $PQ$ edges are sorted by their length, and are added at the threshold $\alpha \geq 0$ corresponding to the length of the edge. This process is visualized in more details on Figures 10 and 11. The triangles are added at the threshold at which the last of its three edges are added. The $3-$ and higher $k-$simplices are added similarly at the threshold corresponding to the adding of the last of their edges. The added triangles and higher dimensional simplices are not shown explicitly on Figure 1 for ease of perception, as they can be restored from their edges. As all simplices within the $Q-$cloud are added at the very beginning at $\alpha = 0$, the corresponding cycles formed by the $Q-$cloud simplices are immediately killed at $\alpha = 0$ and do not contribute to the Cross-Barcode.

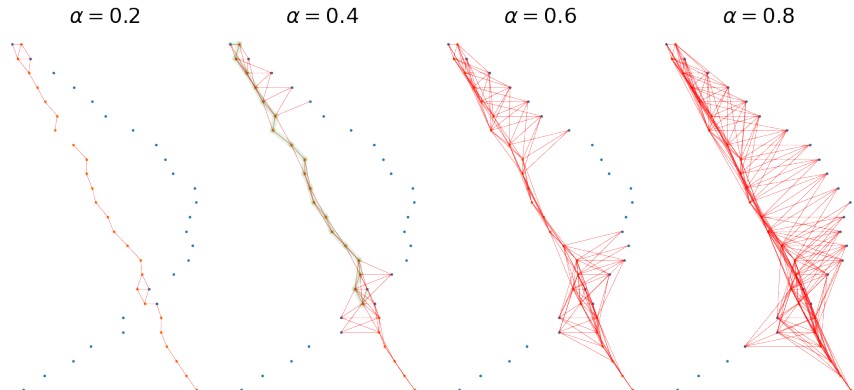

Figure 10: We are adding edges between $P-$points(orange) and $Q-$points(blue) and between pairs of $P-$points consecutively. The edges are sorted by their length, and are added at the threshold $\alpha \geq 0$ corresponding to the length of the edge. Here at the thresholds $\alpha = 0.2, 0.4, 0.6, 0.8$ edges with length less than $\alpha$ were added. For ease of perception the simultaneously added triangles and higher simplices, as well as the added at $\alpha = 0$ all simplices between $Q-$points, are not shown explicitly here. Notice how the $1-$cycle, shown with green, with endpoints in $Q-$cloud is born at $\alpha = 0.4$. It survives at $\alpha = 0.6$ and it is killed at $\alpha = 0.8$.

The constructed set of simplices is naturally a simplicial complex, since for any added $k-$simplex, we have added also all its $(k-1)-$faces obtained by deletion of one of vertices. The threshold $\alpha$ defines the filtration on the obtained simplicial complex, since the simplices added at smaller threshold $\alpha_1$ are contained in the set of simplices added at any bigger threshold $\alpha_2 > \alpha_1$.

With adding more edges, the cycles start to appear. In our case, a cycle is essentially a collection of simplices whose boundary is allowed to be nonzero if the boundary consists of simplices with vertices from $Q$. For example, a $1-$ cycle in our case is a path consisting of added edges, that can start and end in $Q-$cloud and that passes through $P-$points. This is because any such collection can be completed to a collection with zero boundary since any cycle from $Q-$cloud is a boundary of a sum of added at $\alpha = 0$ simplices from $Q$.

A $1-$cycle disappears at the threshold when a set of triangles is added whose boundary coincides with the $1-$cycle plus perhaps some edges between $Q-$points.

Notice how the $1-$cycle with endpoints in $Q-$cloud is born at $\alpha = 0.4$ on Figure 10, shown with green. It survives at $\alpha = 0.6$ and it is killed at $\alpha = 0.8$. The process of adding longer edges can be visually assimilated to the building of a "spider's web" that tries to bring the cloud of red points closer to the cloud of blue points. The obstructions to this are quantified by "lifespans" of cycles, they correspond to the lengths of segments in the barcode. See e.g., Figure 11 where a $1-$cycle is born between $\alpha = 0.5$ and $0.9$, it then corresponds to the green segment in the Cross-Barcode.

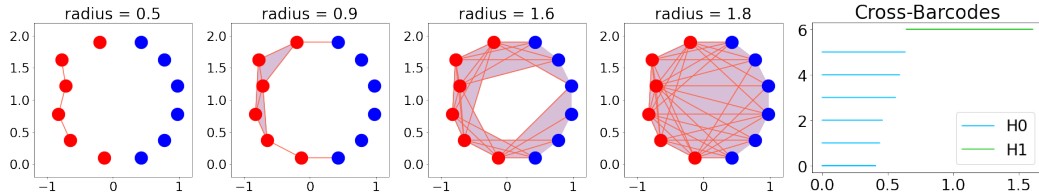

Figure 11: The process of adding the simplices between the $P-$cloud(red) and $Q-$cloud(blue) and within the $P-$cloud. Here we show the consecutive adding of edges together with simultaneous adding of triangles. All the edges and simplices within $Q-$cloud are assumed added at $\alpha = 0$ and are not shown here for perception's ease. Notice the $1-$cycle born between $\alpha = 0.5$ and $0.9$, it corresponds to the green segment in the shown Cross-Barcode

**Remark A.1.** *To characterise the situation of two data point clouds one of which is a subcloud of the other $S \subset C$, it can be tempting to start seeking a "relative homology" analog of the standard (single)*

*point cloud persistent homology. The reader should be warned that the common in the literature "relative persistent homology" concept and its variants, i.e. the persistent homology of the decreasing sequence of factor-complexes of a fixed complex: $K \to \dots K/K_i \to K/K_{i+1} \to \dots K/K$, is irrelevant in the present context. In contrast, our methodology, in particular, does not involve factor-complexes construction, which is generally computationally prohibitive. The point is that the basic concept of filtered complex contains naturally its own relative analogue via the appropriate use of various filtrations.*

### A.3   Sub-manifolds and bars in Cross-Barcode$_*(P, Q)$

It is natural to start analyzing the closeness of the data point cloud $P$ to the data point cloud $Q$ by looking at the matrix of the $PQ$ pairwise distances. If there are many points from $P$ such that their distance to their closest point from $Q$ is relatively big then the clouds $P$ and $Q$ are not close. However, in applications, it is important to distinguish the different situations here. The first case is when all these remote from **Q** points are close to each other. Then this remote from $Q$ cluster of $P$ points represents a single topological feature distinguishing cloud $P$ from $Q$. Another case is when the remote from $Q$ points form several clusters so that each such remote from $Q$ cluster represents a separate topological feature. The long bars in the zero-dimensional Cross-Barcode record the lifespans on the distances' scale of these remote from $Q$ clusters of $P-$points.

In practice it also happens more often that it is not possible to distinguish a separate cluster of $P$ points which are all remote from $Q$. Rather, there are some $P-$points inside the same $P-$cluster that are close to $Q$ and other $P-$points from the same $P-$cluster which are further away from $Q$, as on Fig.1. This situation is captured and quantified by the higher dimensional topological features distinguishing cloud $P$ from $Q$. Intuitively such an $i-$dimensional topological feature represents an $i-$dimensional $P-$cloud's sub-manifold whose boundary is close to the $Q-$cloud, but whose interior $P-$points are remote from the $Q-$cloud, like the green polygonal chain on Fig.10 at $\alpha = 0.4$. Such features are constructed in the algorithm using the distance matrix combinatorics from $(i+1)-$tuples of $P-$points or $P$ & $Q-$ points. The distances within each of these tuples are less or equal to the feature's appearance, or birth, threshold. The disappearance, or death, of such a feature calculated by the algorithm corresponds approximately to the scale at which the feature becomes indistinguishable from the $Q-$cloud. The $i-$dimensional Cross-Barcode$_i(P, Q)$, $i \geq 1$, is the set of segments (bars) recording the birth and the death thresholds of such topological features.

### A.4   Cross-Barcode$_*(P, Q)$ as obstructions to assigning $P$ points to distribution $\mathcal{Q}$

Geometrically, the lowest dimensional Cross-Barcode$_0(P, Q)$ is the record of relative hierarchical clustering of the following form. For a given threshold $r$, let us consider all points of the point cloud $Q$ plus the points of the cloud $P$ lying at a distance less than $r$ from a point of $Q$ as belonging to the single $Q-$cluster. It is natural to form simultaneously other clusters based on the threshold $r$, with the rule that if the distance between two points of $P$ is less than threshold $r$ then they belong to the same cluster. When the threshold $r$ is increased, two or more clusters can collide. And the threshold, at which this happens, corresponds precisely to the "death" time of one or more of the colliding clusters. At the end, for very large $r$ only the unique $Q-$cluster survives. Then Cross-Barcode$_0(P, Q)$ records the survival times for this relative clustering.

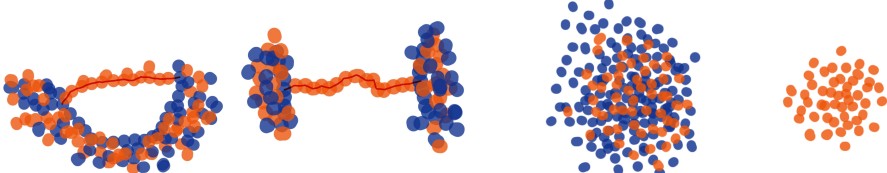

Figure 12: Paths/membranes (red) in the void that are formed by small intersecting disks around $P$ points (orange), and are ending on $Q$ (blue), are obstacles for identification of the distribution $\mathcal{P}$ with $\mathcal{Q}$. These obstacles are quantified by Cross-Barcode$_1(P, Q)$. Separate clusters are the obstacles quantified by Cross-Barcode$_0(P, Q)$.

Notice that in situations, like, for example, in Figure 12, it is difficult to attribute confidently certain points of $P$ to the same distribution as the point cloud $Q$ even when they belong to the "big" $Q-$cluster

at a small threshold $r$, because of the nontrivial topology. Such "membranes" of $P-$points in void space, are obstacles for assigning points from $P$ to distribution $\mathcal{Q}$. These obstacles are quantified by the segments from the higher barcodes Cross-Barcode$_{\geq 1}(P, Q)$. The bigger the length of the associated segment in the barcode, the further the membrane passes away from $Q$.

### A.5 More simple synthetic datasets in 2D

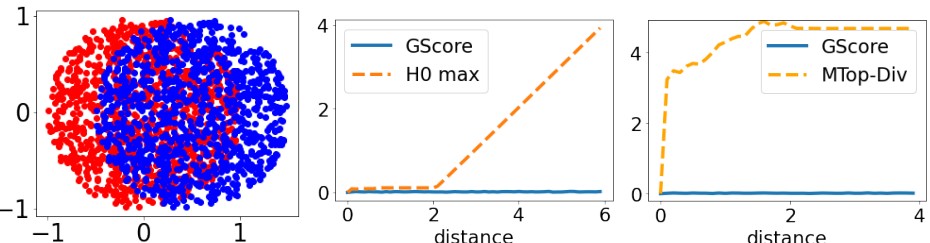

Figure 13: The first picture shows two clouds of 1000 points sampled from the uniform distributions on two different disks of radius 1 with a distance between the centers of the disks of 0.5. The second and third pictures show the dependence of the GScore metric (the GScore is equal to zero independently of the distance between the disks), maximum of segments in $H0$ and the sum of lengths of segments H1 as a function of the distance between the centers of the disks averaged over 10 runs. The length of the maximum segment barcode in H0 grows linearly and equals to the distance between the pair of closest points in the two distributions.

As illustrated on Figures 2,13 the GScore is unresponsive to changes of the distributions' positions.

### A.6 Cross-Barcode and precision-recall

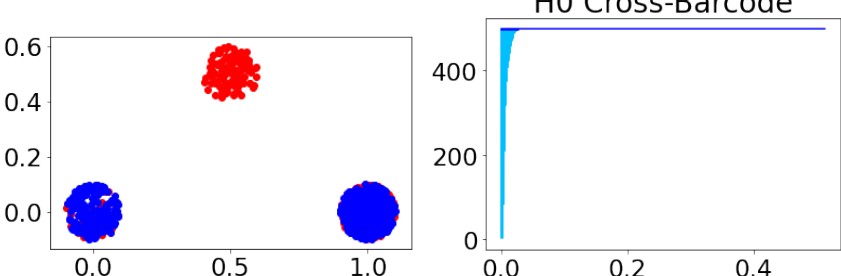

Figure 14: Mode-dropping, bad recall & good precision, illustrated with clouds $P_{\text{data}}$ (red) and $Q_{\text{model}}$ (blue). The Cross-Barcode$_0(P_{\text{data}}, Q_{\text{model}})$ contains long intervals, one for each dropped mode, which measure the distance from the data's dropped mode to the closest generated mode.

The Cross-Barcode captures well the precision vs. recall aspects of the point cloud's approximations, contrary to FID, which is known to mix the two aspects. For example, in the case of mode-dropping, bad recall but good precision, the Cross-Barcode$_0(P_{\text{data}}, Q_{\text{model}})$ contains the long intervals, one for each dropped mode, which measure the distance from the data's dropped mode to the closest generated mode. The mode-dropping case (bad recall, good precision) is illustrated on Figure 14.

Analogously, in the case of mode-invention, with good recall but bad precision, the Cross-Barcode$_0(Q_{\text{model}}, P_{\text{data}})$ contains long intervals, one for each invented mode, which measure the distance from the model's invented mode to the closest data's mode.

The mode-invention (good recall, bad precision) case is illustrated on Figure 15.

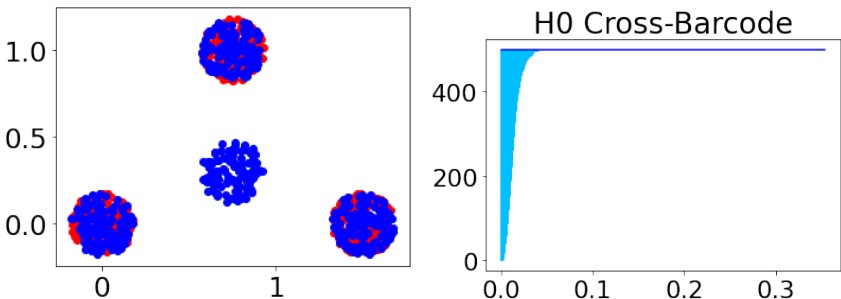

Figure 15: Mode-invention, good recall & bad precision, illustrated with clouds $P_{\text{data}}$ (red) and $Q_{\text{model}}$ (blue). The Cross-Barcode$_0(Q_{\text{model}}, P_{\text{data}})$ contains long intervals, one for each invented mode, which measure the distance from the model's invented mode to the closest data's mode.

## B Cross-Barcode Properties.

### B.1 The Cross-Barcode's norm and the Hausdorf distance.

The Bottleneck distance [11], also known as Wasserstein$-\infty$ distance $\mathbb{W}_\infty$, defines the natural norm on the Cross-Barcodes:

$$\|\text{Cross-Barcode}_i\,(P,Q)\|_B = \max_{[b_j, d_j] \in \text{Cross-Barcode}_i} (d_j - b_j).$$

The Hausdorf distance measures how far are two subsets $P, Q$ of a metric space from each other. The Hausdorf distance is the greatest of all the distances from a point in one set to the closest point in the other set:

$$d_{\text{H}}(P, Q) = \max \left\{ \sup_{x \in P} d(x, Q), \sup_{y \in Q} d(y, P) \right\}. \tag{4}$$

**Proposition 1.** *The norm of Cross-Barcode$_i(P,Q)$, $i \geq 0$, is bounded from above by the Hausdorff distance*

$$\|\text{Cross-Barcode}_i\,(P,Q)\|_B \leq d_H(P,Q). \tag{5}$$

*Proof.* Let $c \in R_{\alpha_0}(\Gamma_{P \cup Q}, m_{(P \cup Q)/Q})$ be an $i-$dimensional cycle appearing in the filtered complex at $\alpha = \alpha_0$. Let us construct a simplicial chain that kills $c$. Let $\sigma = \{x_1, \ldots, x_{i+1}\}$ be one of the simplices from $c$. Let $q_j$ denote the closest point in $Q$ to the vertex $x_j$. The prism $\{x_1, q_1, \ldots, x_{i+1}, q_{i+1}\}$ can be decomposed into $(i+1)$ simplices $p_k(\sigma) = \{x_1, x_2, \ldots, x_{k-1}, q_k, \ldots, q_{i+1}\}, 1 \leq k \leq i+1$. The boundary of the prism consists of the two simplices $\sigma$, $q(\sigma) = \{q_1, \ldots, q_{i+1}\}$, and of the $(i+1)$ similar prisms corresponding to the the boundary simplices of $\sigma$. If $c = \sum_n a_n \sigma^n$ then

$$c = \partial(\sum_n a_n \sum_k p_k(\sigma^n)) + \sum_n a_n q(\sigma_n)$$

For any $k, j$, $d(x_j, x_k) \leq \alpha_0$ since $c$ is born at $\alpha_0$. Therefore

$$d(x_j, q_k) \leq d(x_j, x_k) + d(x_k, q_k) \leq \alpha_0 + \sup_{x \in P} d(x, Q).$$

Therefore all simplices $p_k(\sigma^n))$ appear no later than at $(\alpha_0 + \sup_{x \in P} d(x, Q))$ in the filtered complex. All vertices of the simplices $q(\sigma_n)$ are from $Q$. It follows that the lifespan of the cycle $c$ is no bigger than $\sup_{x \in P} d(x, Q))$ $\qquad \square$

To illustrate the proposition 1 we have verified empirically the diminishing of Cross-Barcode$_*(Q_1, Q_2)$ when number of points in $Q_1$, $Q_2$ goes to $+\infty$ and $Q_1$, $Q_2$ are sampled from the same uniform distribution on the 2D disk of radius 1. The maximal length of segments in $H1$ as function of number of points in the clouds of the same size is shown in Figure 16.

## B.2 MTop-Div and the Cross-Barcode's Relative Living Times (RLT)

The Cross-Barcode for a given homology $H_i$ is a list of birth-death pairs (segments)

$$\text{Cross-Barcode}_i(P, Q) = \{[b_j, d_j]\}_{j=1}^n$$

Relative Living Times is a discrete distribution $RLT(k)$ over non-negative integers $k \in \{0, 1, \ldots, +\infty\}$. For a given $\alpha_{max} > 0$, $RLT(k)$ is a fraction of "time", that is, parts of horizontal axis $\tau \in [0, \alpha_{max}]$, such that exactly $k$ segments $[b_i, d_i]$ include $\tau$.

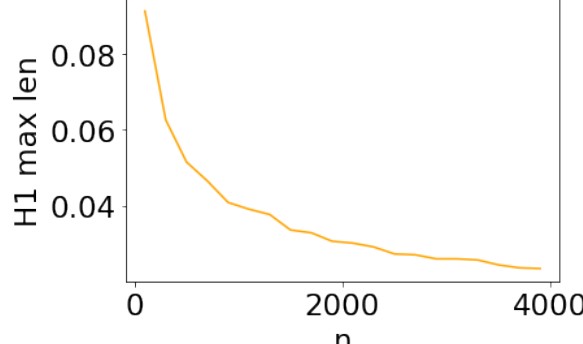

Figure 16: Diminishing of the length of maximal segment in $H_1$ with increase of the number of sampled points for point clouds of the equal size $n$ sampled from the same uniform distribution on the 2D disk of radius 1.

For equal point clouds, Cross-Barcode$_i(P, P)$ = $\varnothing$ and the corresponding RLT is the discrete distribution concentrated at zero. Let us denote by $O_0$ such discrete distribution corresponding to the empty set. A natural measure of closeness of the distribution RLT to the distribution $O_0$ is the earth-mover's distance (EMD), also called Wasserstein-1 distance.

**Proposition**. Let for all $d_i \leq \alpha_{max}$, then

$$\text{MTop-Div}(P, Q) = \alpha_{max}\text{EMD}(RLT(k), O_0).$$

*Proof.* By the definition of EMD

$$\text{EMD}(RLT, O_0) = \sum_{k=1}^{+\infty} k \times RLT(k).$$

Let's use all the distinct $b_i, d_i$ to split $[0, \alpha_{max}]$ to disjoint segments $s_j$:

$$[0, \alpha_{max}] = \bigsqcup_j s_j.$$

Each $s_j$ is included in $K(j)$ segments $[b_i, d_i]$ from the Cross-Barcode$_i(P, Q)$. Thus,

$$RLT(k) = \frac{1}{\alpha_{max}} \sum_{j:K(j)=k} |s_j|.$$

At the same time:

$$\text{MTop-Div}(P, Q) = \sum_i (d_i - b_i) = \sum_j K(j)|s_j| = \sum_{k=1}^{+\infty} \sum_{j:K(j)=k} K(j)|s_j|$$

$$= \sum_{k=1}^{+\infty} \alpha_{max} \times k \times RLT(k) = \alpha_{max}\text{EMD}(RLT(k), O_0).$$

$\square$

## C  Hyperparameters, Software used, and Experiments' Details

We have made experiments in various settings and on the following datasets:

- on a set of gaussians in 2D in comparison with distributions generated by GAN and WGAN.
- **MNIST** We have observed that GScore is not sensitive to the flip of the cloud of "fives", while our score MTop-Divergence is sensitive to such flip since it depends on the positions of clouds with respect to each other

- **CIFAR10** We have evaluated our MTop-Div(D,M) using a benchmark with the controllable disturbance level. We have observed that Geometry Score is monotone only for 'mode invention' and 'intra-mode collapse' while MTop-Div(D,M) is almost monotone for all the cases.

- **FFHQ** We have evaluated the quality of distributions generated by StyleGAN and StyleGAN-2, without truncation and with $\psi = 0.7$ truncation. We observed that the ranking via MTop-Div is consistent with FID

- **ShapeNet**[6] We have studied the training dynamics of the GAN trained on 3D shapes. We observed that MTop-Div is consistent with domain specific measures (JSD, MMD, Coverage) and that MTop-Div better describes the evolution of the point cloud of generated objects during epochs;

- **Stock data** We have studied the training dynamics of TimeGAN [7] applied to market stock data. We observed that MTop-Div is consistent with the discriminative score but better captures the evolution of point cloud of generated objects during epochs;

- **Chest X-ray images** We have studied the training dynamics of ACGAN applied to chest X-ray images. We observed that MTop-Div is more consistent with the discriminative score than FID;

For computation of FID we used Pytorch-FID[8]. For computation of Geometry Score we used the original code[9] patched to supported multi-threading, otherwise it was extremely slow. The RLTs computation was averaged over 2500 trials. We calculated persistent homology via ripser++[10].

We used the following hyperparameters to compute MTop-Div:

- MNIST: $b_P = 10^2, b_Q = 10^3$;

- Gaussians: $b_P = 10^2, b_Q = 10^3$;

- CIFAR10: $b_P = 10^3, b_Q = 10^4$;

- FFHQ: $b_P = 10^3, b_Q = 10^4$.

- ShapeNet: $b_P = 10^2, b_Q = 10^3$;

- Market stock data: $b_P = 10^2, b_Q = 10^3$;

- Chest X-ray data: $b_P = 10^2, b_Q = 10^3$.

MTop-Div scores were were averaged over 20 runs.

We compared Geometry Score and MTop-Div in the experiment with mixtures of Gaussians. Table 4 shows the results. We conclude that the MTop-Div is consistent with the visual quality of GAN's output while Geometry Score fails.

Figure 17 shows Cross-Barcodes for the experiment with StyleGAN's trained on FFHQ. Figure 19 shows one of Cross-Barcodes in $H_0$ from the experiment with CIFAR10 dataset to illustrate that the $0-$dimensional Cross-Barcode can also be applied. Figure 18 shows the Cross-Barcodes in $H_1$ from the experiments with GAN[11] and WGAN-GP [12] trained on mixture of Gaussians.

Table 3 shows extended experimental results on GAN model selection including standard error of sample means of MTop-Div.

Figure 20 presents real and generated chest X-ray images. The generated images are of high visual quality and resembles real images.

Figure 21 shows real and generated 3D shapes. Generated 3D shapes (bottom row) are relatively blurry.

---

[6]The dataset is free for non-commercial purposes.

[7]https://github.com/jsyoon0823/TimeGAN

[8]https://github.com/mseitzer/pytorch-fid, (Apache Licence 2.0)

[9]https://github.com/KhrulkovV/geometry-score

[10]https://github.com/simonzhang00/ripser-plusplus, (MIT License)

[11]https://arxiv.org/abs/1406.2661

[12]https://arxiv.org/abs/1704.00028

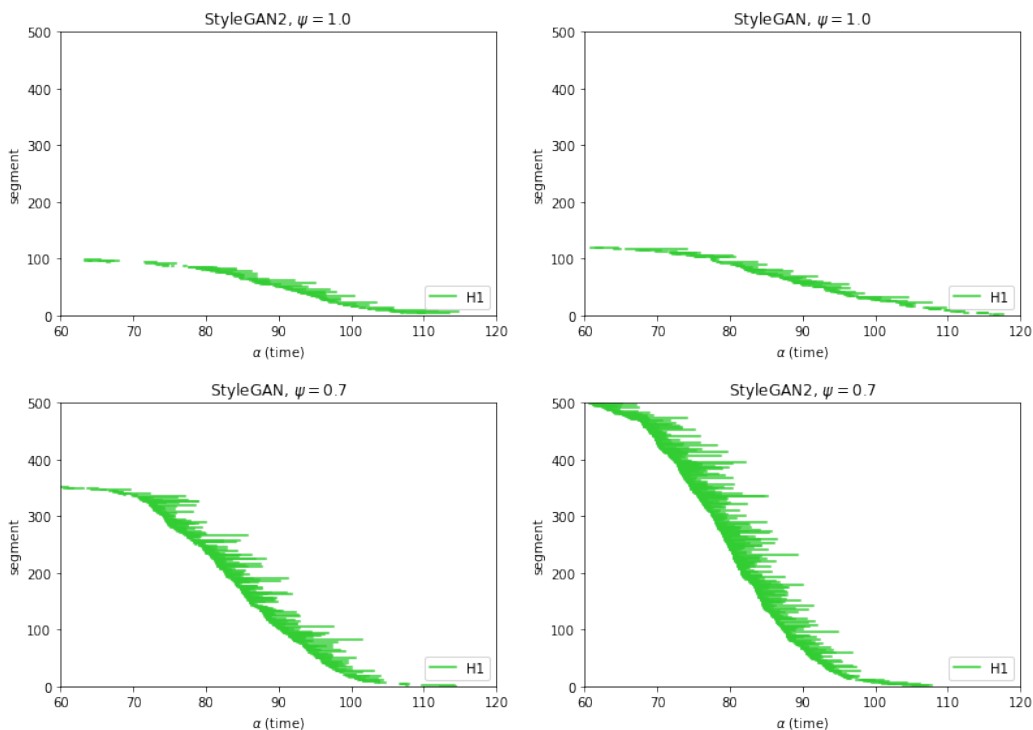

Figure 17: Cross-Barcodes for GAN's trained on FFHQ. Cross-Barcodes are shown by decrease of generator performance. For clarity, only $H_1$ barcodes are shown. The number and the total length of segments give the same ranking as the FID score

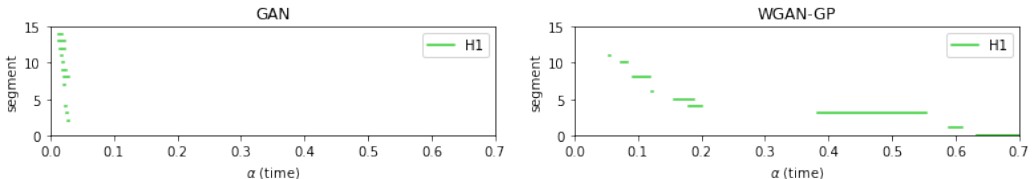

Figure 18: Cross-Barcodes for GAN's trained of mixtures of Gaussians. Cross-Barcodes are shown by decrease of generator performance. For clarity, only $H_1$ barcodes are shown.

## D MTop-Div for Cross-Barcodes of higher order

We calculated MTop-Div$_k$(D,M) based on higher order Cross-Barcodes, that is, sums of segments' lengths of Cross-Barcode$_k$, $k > 1$ were taken in Algorithm 2. Then, we measured average Kendall-tau rank correlation between MTop-Div$_k$(D,M) and the disturbance level for the series of synthetic modifications of CIFAR10. For MTop-Div$_2$(D,M) the rank correlation was 0.59, for MTop-Div$_3$(D,M): 0.45. To make faster calculations small batches were selected, MTop-Div$_2$(D,M): $b_P = 100, b_Q = 300$, MTop-Div$_3$(D,M): $b_P = 100, b_Q = 100$. An optimization that we describe in a future publication pre-computes the unnecessary simplices and permits faster higher degree MTop-Div computations.

Table 2: Performance measures of StyleGANs trained on FFHQ.

| GAN | $\psi$ | FID | MTop-Div(m,d) |
|---|---|---|---|
| STYLEGAN2 | 1.0 | 4.75 | 162.08 |
| STYLEGAN | 1.0 | 8.25 | 234.33 |
| STYLEGAN | 0.7 | 15.86 | 712.57 |
| STYLEGAN2 | 0.7 | 19.75 | 1011.53 |

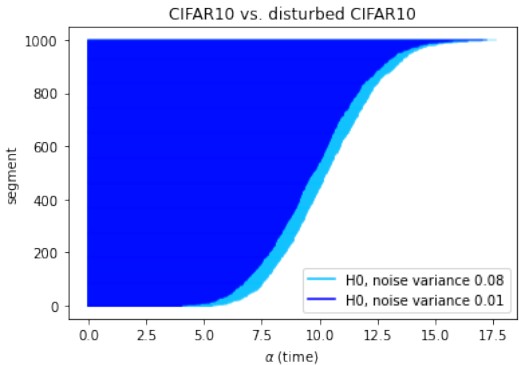

Figure 19: Cross-Barcodes for CIFAR10 vs. disturbed CIFAR10. Gaussian noise with 2 levels of variance was applied. For clarity, only $H_0$ barcodes are shown. For ease of perception of differences in Cross-Barcodes$_0$ they are shown on the same plot. The dataset with higher level of noise is distinguished here by the longer segments in the Cross-Barcode$_0$

Table 3: MTop-Div is consistent with FID for model selection of GAN's trained on various datasets.

| Dataset | FID | | MTop-Div(M,D) | |
|---|---|---|---|---|
| | WGAN | WGAN-GP | WGAN | WGAN-GP |
| CIFAR10 | **154.6** | 399.2 | **370.5±17.3** | 2408.5±27.0 |
| SVHN | **101.6** | 154.7 | **332.0±12.4** | 963.2±22.62 |
| MNIST | 31.8 | **22.0** | 2365.6±40.1 | **1474.2±29.7** |
| FashionMNIST | 52.9 | **35.1** | 1052.6±24.8 | **872.9±21.8** |

# E    Comparison with the "Intrinsic Multi-scale Distance(IMD)"

As proposed by a reviewer, we did additional experiments with IMD [31] applied to point clouds from our experiments. IMD is not sensitive to the rings shift (Section 3.1) and the digits flipping on MNIST (Section 3.3). For the experiment "Mode dropping on Gaussians" (Section 3.2), IMD incorrectly ranks poorly performing WGAN-GP (see Fig.3) higher than the original GAN (Table 4). For the experiments "GAN model selection" (Section 3.5), IMD ranks a better performing model lower in one case, while the ranking via MTop-Div is consistent with true GAN performance. For the "Synthetic variations of CIFAR10" experiment (Section 3.4), the average Kendall-tau correlation between IMD score and the disturbance level is 0.55, which is lower than the same measure of MTop-Div (0.89).

Table 4: MTop-Div and G. Score for GAN's trained of mixtures of Gaussians.

| GAN | G. SCORE | MTOP-DIV(M,D) | MTOP-DIV(D,M) | IMD |
|---|---|---|---|---|
| WGAN-GP | **1.083** | 0.562 | 0.206 | **2.65** |
| ORIG. GAN | 1.087 | **0.081** | **0.149** | 13.87 |

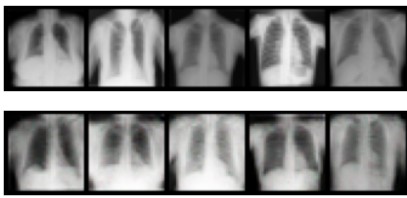

Figure 20: Top: real chest X-ray images. Bottom: generated chest X-ray images.

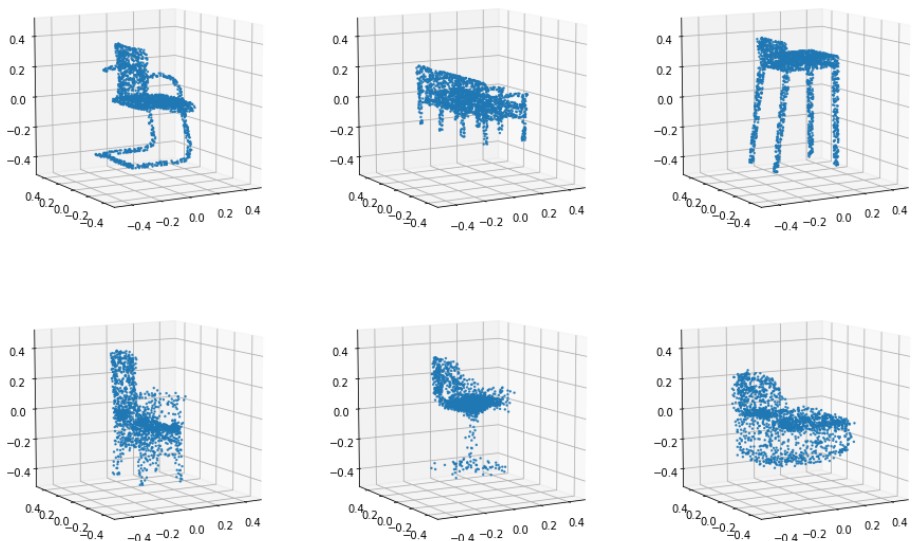

Figure 21: Top: real 3D shapes. Bottom: generated 3D shapes. Generated 3D shapes are relatively blurry.