# OpenReview forum: "Manifold Topology Divergence: a Framework for Comparing Data Manifolds. "
_NeurIPS.cc/2021/Conference — NeurIPS 2021 Poster_

### Official Review · Reviewer_8Pta · 2021-07-14

**Rating:** 6
**Confidence:** 3

**Summary:**

The authors propose a method to compare two data manifolds based on topology. A persistent diagram is constructed using both sets of samples (cross-barcode), and the proposed metric is computed as the average length of segments in the cross-barcode. The efficiency of the method is demonstrated in several experimental settings.

**Limitations And Societal Impact:**

The authors do not discuss potential negative societal impact, neither the limitations of their work.

**Main Review:**

Originality:
First of all, I am not an expert in topological data analysis. However, it seems that the cross-barcode is new idea, which is shown to be potentially useful. Also, I think that the proposed idea is a sufficient contribution compared to the most related method (Geometry Score). As regards the related work, I believe that the paper "The Shape of Data: Intrinsic Distance for Data Distributions", A. Tsitsulin, et. al., ICLR 2020, should be cited and probably included in the experiments.


Quality:
As far as I can tell, the technical part of the paper seems to be correct. In addition, the claims are supported by the experiments. Maybe some toy data with nonlinear manifolds could be used in order to show the behavior in complicated scenarios. The authors do not mention any weakness of the methodology, and I could not think of such a case (see Clarity).


Clarity:
In my opinion, the clarity of the paper should be improved. In particular, it is quite difficult for a non-expert reader (like me) to understand the intuition behind the proposed approach. Especially, in Sec 2.2 - 2.6 there is a lot of text that is not easily accessible. I think that some images could help the reader to understand better the proposed idea. Also, I am aware that the used terminology is in accordance to the topological data analysis field, but instead, I would like the authors to provide a manuscript easier to understand from a non-expert reader. Anyways, the proposed method aims researches in machine learning that potentially need a tool to compare generative models, and not only researchers familiar with topology.


Significance:
I think that the proposed idea is potentially quite useful, since it provides a tool to assess the performance of generative modelling. This is, in general, rather useful for machine learning applications. From the experiments, it seems that the method is efficient and better from the related models. However, my main concern is that as a non-expert reader, I cannot understand in depth the proposed idea, so probably I cannot judge it objectively and find the potential weaknesses. In my opinion this limits the significance of the method.

**Time Spent Reviewing:**

6

---

> ### Author Response · Authors · 2021-08-09
> **Response to the Review #3**
>
> Thank you for your time and thorough review.  We will improve the presentation according to the suggestions. Below we address specific concerns one by one.
>
> __Q1__:  _As regards the related work, I believe that the paper "The Shape of Data: Intrinsic Distance for Data Distributions", A. Tsitsulin, et. al., ICLR 2020, should be cited and probably included in the experiments._ \
> __A__:
> Thank you for drawing our attention to the paper "The Shape of Data: Intrinsic Distance for Data Distributions", A. Tsitsulin, et. al., ICLR 2020”.
> We have calculated the IMD scores for the point clouds from our experiments:
>
> IMD is not sensitive to the rings shift (section 3.1) and the digits flipping on MNIST (section 3.3).
> For the experiment “Mode dropping on Gaussians” (section 3.2), IMD incorrectly ranks poorly performing WGAN-GP (see Fig.3) higher than the original GAN:
>
> |   |  IMD | GScore |  MTopDiv |
> |---|:---:|:---:|:---:|
> | WGAN-GP | __2.65__ |1.083 |0.562|
> | orig. GAN |13.87|1.087|__0.081__|
>
> For the experiments “GAN model selection” (section 3.5), the IMD score ranks a better performing model lower in one case, while the ranking via MTop-Div is consistent with performance.
> For the “Synthetic variations of CIFAR10” experiment (section 3.4), the average Kendall-tau correlation between IMD score and the disturbance level is 0.55 which is lower than the same measure of MTop-Div (0.89).
> We are adding the citation and these comments to the paper.
>
> __Q2__: _In particular, it is quite difficult for a non-expert reader (like me) to understand the intuition behind the proposed approach._ \
> __A__: Thanks for the feedback. We are adding more intuition and explanations aimed in particular at a non-expert reader.
>
> Due to the space limitations, we had to omit a detailed exposition of the homology and simplicial complexes technique in the paper's main body and to give all the necessary definitions in the appendix while referring to the established textbooks for a thorough account. Here is yet another intuitive point of view introducing the Cross-Barcode:
>
> It is natural to start analyzing the closedness of the data point cloud P to the data point cloud Q by looking at the matrix of the P-Q pairwise distances. If there are many points $p_i$ from P such that their distance to their closest point from Q  is relatively big then the clouds P and Q are not close. However, in applications, it is important to distinguish the different situations here. The first case is when all these remote from Q points $p_i$ are close to each other. Then this remote from Q cluster of P points represents a single topological feature distinguishing cloud P from Q.  Another case is when the remote from Q points $p_i$ form several clusters so that each such remote from Q cluster represents a separate topological feature. The long bars in the zero-dimensional  Cross-Barcode$_0(P,Q)$ record the lifespans on the distances’ scale of these remote from Q clusters of P-points.
>
> In practice it also happens more often that it is not possible to distinguish a separate cluster of P points which are all remote from Q. Rather,  there are some P points inside the same P cluster  that are close to Q and other P points from the same P cluster which are further away from Q, as on Fig.1. This situation is captured and quantified by the higher dimensional topological features distinguishing cloud P from Q. Intuitively such an i-dimensional topological feature represents an i-dimensional P-cloud's submanifold whose boundary is close to the Q-cloud, but whose interior P-points are remote from the Q-cloud, like the green polygonal chain on Fig.10 (appendix) at $\alpha=0.4$.  Such features are constructed in the algorithm using the distance matrix combinatorics from (i+1)-tuples of P points or P & Q points. The distances within these tuples are less or equal to the birth threshold, where the birth thresholds of the features are also calculated by the algorithm.  The death of such a feature calculated by the algorithm corresponds approximately to the scale at which the feature becomes indistinguishable from the Q cloud.
> The i-dimensional $\text{Cross-Barcode}_i(P,Q)$, $i\geq 1$, is the set of bars recording the birth and the death thresholds of such topological features.
>
> The construction of such features is also described via sequential adding of edges, triangles, 3-simplexes, and so on, constituting the cycles showing the discrepancies between two clouds, on lines 429-460, see also Fig.10-11(appendix).
>
> We are adding a comment on this intuitive point of view and an extra illustration to the paper.
>
> __Q3__: _Maybe some toy data with nonlinear manifolds could be used in order to show the behavior in complicated scenarios. I think that some images could help the reader to understand better the proposed idea._
> __A__: Thanks for these remarks. Some toy data examples with nonlinear manifolds are presented in Fig.2 (two rings and their shifts) in section 3.1, and in Fig.10,11,13 in the appendix.  We will add more links to these examples, in particular, in section 2.3. The behavior of the Cross-Barcode$_1$ having the two long bars on Fig.2, center left, is quite instructive. The two longer bars represent the topological features given by two blue (P) points' circular segments whose boundaries lie on the intersection with the red (Q) points' cloud. The lengths of the two long bars in the Cross-Barcode$_1$ coincide essentially with the maximal distances to the closest red point for blue points from the two circular segments. With further increase of the distance $d$ between the rings, first both long bars in the Cross-Barcode$_1$ increase, then one of the two bars continues to increase while the second decreases. We are also adding another example to the final version.
>
>
> __Q4__: _the proposed metric is computed as the average length of segments in the cross-barcode_ \
> __A__: The MTop-Div is the average _sum of lengths_ of segments in the Cross-Barcode$_1$. It incorporates both the average length of segments and the total number of segments.
>
> __Q5__: _The authors do not mention any weakness of the methodology_
> __A__: Thanks for this remark. The limitations of the method are mentioned briefly, in particular, in the conclusions: we have tested only datasets having up to  $10^5$ samples and the dimension up to $10^7$; limitations implied by the algorithm's complexity are given in Section 2.6; computations of higher-order barcodes may be more expensive than the first order. We will discuss this in more depth in the final version.
>
>
> Concluding remarks. Please respond to our post to let us know if the clarifications above suitably address your concerns about our work. We are happy to address any remaining points during the discussion phase; if the responses above are sufficient, we kindly ask that you consider raising your score.

---

> > ### Comment · Reviewer_8Pta · 2021-08-24
> > **After rebuttal**
> >
> > As I am not an expert in TDA, I got influenced by the other reviewers about the significance of the submission. Also, the authors tried to answer my questions, and in addition, they are willing to take my clarity comments into account. I will increase my score to 6 and vote for acceptance.

---

### Official Review · Reviewer_ABSW · 2021-07-15

**Rating:** 9
**Confidence:** 4

**Summary:**

This work introduces a novel method based on relative persistent homology (because the term of persistent relative homology has been occupied before, referring to completely different concept) for measuring the similarity between manifolds, especially the sample point clouds of the manifolds. The distance uses the bar-code of persistent homology, and the homology groups are relative groups H_k(P,Q). The relative between the cross-bar codes and the Hausdorff distance between the point clouds   is proved in the appendix.  Based on the cross-barcode, the manifold topology divergence score (MTop-Divergence) is introduced and applied to assess generative models in various domains. The MTop-Div can detect mode collapsing , mode invention more effectively.  This is a first TDA-based practical method, and has many merits, scalability, domain agnostic, independence of pre-trained networks.

**Limitations And Societal Impact:**

The  persistent homology is constructed using the Euclidean distance, for manifold data analysis, it is natural to consider geodesic distance.

The method focuses on the topology of the simplicial complex, most of the times, the regularity of the manifold also plays an important role. The Hausdorff distance is not sufficient to ensure the convergence of manifolds with Riemannian metrics.





**Main Review:**

This work uses topological data analysis (TDA) method for deep learning, especially for measuring the distance between point clouds, therefore it is suitable for studying the similarities between distributions on manifolds. From theoretical point of view, the work is novel and inspiring, it shows the rigor and elegance of algebraic topology for data analysis. From practical point of view, the work has been tested on various generative models on different domains, the experiments show the effectiveness for discovering mode dropping, collapsing and mode invention. The way of combing powerful applications with solid theories should be promoted in deep learning.

The manuscript is well written. The introduction is highly motivated, and explains the key ideas clearly. The previous work part focuses on the literature in deep learning, and completely omits the large literature for persistent homology. It will be helpful to give some important references in TDA there.

The theoretic section 2 represents the key theoretic results and the idea of the methodology. It is challenging to explain the whole picture to general audience without the knowledge of persistent homology. The authors  give many intuitive examples to explain the idea, and postpone the formulation to the later part, and give formal definitions and prove the claims in the appendix.

The "spider web" metaphor is very intuitive and interesting. The proof for the position 1,  the upper bound of cross-barcode is given by the Hausdorff distance, is convincing. There are some theoretical considerations.
1. All the distances among the points are using the Euclidean metric of the embedding space. Since the distributions are defined on the data manifolds, it is natural to consider the Riemannian metric of the manifolds. It is possible that the geodesic distance is very big but the Euclidean distance is very small. How to address this in the current framework ?

2. It is well known in geometric field that Hausdorff distance is the most preliminary distance to measure the shape similarities.  There are many cases that a sequence of shapes converge to a target shape in Hausdorff distance, but the curvature, area diverge. A more refined method may need to considered.

3. In order to measure the distance between distributions, the most common metric is the Wasserstein distance under Lp norm. The proposed method is sensitive to the topological changes,  therefore easily capture the multiple modes; but optimal transport map emphasizes more on distances. Some quantitative comparisons will be valuable.

The experiments are thorough and convincing, it might be helpful to compare with the methods based on optimal transportation.

In summary, the work introduces rigorous persistent homology method for measuring the distance between distributions defined on manifolds embedded in Euclidean spaces. The work combines mathematical rigor with practical usage. The manuscript is well written, and work is inspiring. I advocate the acceptance.


**Time Spent Reviewing:**

3

---

> ### Author Response · Authors · 2021-08-09
> **Response to the Review #2**
>
> We are grateful for your insightful comments and high evaluation of our work.
>
> __Q__: _It will be helpful to give some important references in TDA there._
> __A__: Thanks for the remark. We will add more references on the TDA literature in the final version.
>
> __Q1__: _The geodesic distance and the Euclidian distance._
> __A__: Thanks for the thoughtful comment. The comparison of the data manifolds approximated using _the geodesic distance_ and _the Euclidean distance_ can be done using a variant of the cross-barcode which compares different distance-like structures on the same data cloud. We will add a remark on this in the paper,  the details will appear in an upcoming work.
>
> __Q2-3__: _The refinements of the Hausdorf distance. The Wasserstein distance._
> __A__: Thanks for the inspiring suggestion. The relations with refinements of _the Hausdorf distance_ and with _the Wasserstein distances_ are exciting directions for future research. We expect Whitehead-type theorems, incorporating distance functions and persistence, to be helpful in this setting.

---

### Official Review · Reviewer_kTLP · 2021-07-16

**Rating:** 7
**Confidence:** 3

**Summary:**

The manuscript introduces the cross-barcode between two point clouds, and uses it to construct the manifold topology divergence (MTD). It is proposed that this can be used for performance evaluation of GANs. A selection of experiments are run to show that the MTD is robust where other evaluation metrics fail.

**Limitations And Societal Impact:**

It is unclear to me how exactly the cross-barcode is constructed. In section 2.3 it says that the simplicial complex is constructed from the metric space consisting of P and Q where the pairwise distances in Q are set to zero. As this space is not Hausdorff, or a metric space, this looks slightly odd at first, and could probably be explained better.

**Main Review:**

It is a shame that most of the cross-barcode is thrown away in favour of just the 1st order one in the construction of the MTD, I assume that this is because computation of the higher order ones are infeasible for large data sets, but it would be nice with some comparisons on smaller sets or at least some more commentary on it.

As a domain agnostic tool I think it has applications, and the construction of the cross-barcode is interesting and could be the object of further research.

**Time Spent Reviewing:**

5

---

> ### Author Response · Authors · 2021-08-09
> **Response to the Review #1**
>
> Thank you for the positive feedback and thoughtful comments.
>
>
> __Q1__: _Higher order MTD_
> __A__: It is still feasible to calculate several higher MTD on interesting datasets.  We will add a remark on this and an example with a calculation of cross-barcode and MTD_2,  MTD_3 and MTD_4 in the final version.
>
> __Q2__: _It is unclear to me how exactly the cross-barcode is constructed. In section 2.3 it says that the simplicial complex is constructed from the metric space consisting of P and Q where the pairwise distances in Q are set to zero._
> __A__: One way to rephrase this is to say that the simplicial complex is constructed starting from the __weighted graph with the distance-like weights__ on edges. And then the persistence barcode is calculated from the filtered simplicial complex of the weighted graph.
> The details on the birth and the death of cycles in this filtered complex are also specified on lines 447-454, see  Fig.1 (section 2.3), Fig.10,11 (appendix).
> We are adding more details on this and are also expanding the related comment on lines 423-460 in the paper.

---

### Decision · Program_Chairs · 2021-09-27

**Decision:**

Accept (Poster)

**Comment:**

Congratulations, the paper is accepted to NeurIPS 2021!
Please make an honest effort making this paper more accessible to general ML audience (non-experts in TDA). Clarify the barcode construction. Include persistent homology literature background. Elaborate limitations. Please include other clarifications, edits and additions as discussed in rebuttal and reviews.